# SynthMap: a generative model for synthesis of 3D datasets for quantitative MRI parameter mapping of myelin water fraction

**Serge Didenko Vasylechko**[1,2]          SERGE.VASYLECHKO@CHILDRENS.HARVARD.EDU
**Simon K. Warfield** [1,2]
**Sila Kurugol** [*1,2]
**Onur Afacan**[†1,2]
[1] *Computational Radiology Laboratory, Boston Children's Hospital, Boston, MA, USA*
[2] *Harvard Medical School, Boston, MA, USA*

**Editors:** Under Review for MIDL 2022

## Abstract

We present a generative model for synthesis of large scale 3D datasets for quantitative MRI parameter mapping of myelin water fraction (MWF). Training robust neural networks for estimation of quantitative MRI parameters requires large amounts of data. Conventional approaches to tackling data scarcity use spatial augmentations, which may not capture a broad range of possible variations when only a very small initial dataset is available. Furthermore, conventional non linear least squares (NNLS) based methods for MWF estimation are highly sensitive to noise, which means that high quality ground truth MWF parameters are not available for supervised training. Instead of using the noisy NNLS based estimates of MWF parameters from limited real data, we propose to leverage the biophysical model that describes how the MRI signals arise from the underlying tissue parameters to synthetically generate a wide variety of high quality data of the corresponding signals and corresponding parameters for training any CNN based architecture. Our model samples parameter values from a range of naturally occurring prior values for each tissue type. To capture spatial variation, the generative signal decay model is combined with a generative spatial model conditioned on generic tissue segmentations. We demonstrate that our synthetically trained neural network provides superior accuracy over conventional NNLS based methods under the constraints of naturally occuring noise as well as on synthetic low SNR images. Our source code is available at: https://github.com/sergeicu/synthmap

**Keywords:** synthetic data, generative models, quantitative mapping, myelin, unsupervised training

## 1. Introduction

Quantitative MRI parameter mapping provides access to the underlying tissue types and biological processes. Multi-component T2 signal decay can be used to determine the myelin bound water content (Alonso-Ortiz et al., 2015), which is crucial to the understanding of tissue maturation, disease and injury in many neurological disorders such as multiple sclerosis, Alzheimer's and schizophrenia (Davis et al., 2003; Nasrabady et al., 2018; Popescu and Lucchinetti, 2012). However, estimation of the parameters that govern multi-component

---

[*] Contributed equally

[†] Contributed equally

T2 signal decay is a challenging ill-posed inverse problem. Conventional NNLS based optimization schemes are highly sensitive to noise and typically exhibit poor repeatability and precision (Canales-Rodríguez et al., 2021). This limits their use in clinical practice. From the deep learning perspective, this also limits their potential for supervision during training, since these conventional methods' estimates cannot be reliably used as the ground truth. Furthermore, conventional myelin mapping acquisition sequences are not routinely acquired in clinical scans due to their substantial long acquisition times (Akhondi-Asl et al., 2016), which further limits the availability of sufficient datasets that could be used for training any 2D or 3D based neural network. As a result, training robust and generalizable deep learning models in this kind of setting becomes increasingly difficult.

A conventional approach to limited training data may use spatial augmentations via simple image transforms, which leads to some limited improvements in performance(Chaitanya et al., 2019). A breadth of works also exist in the transfer learning domain, whereby a model is initialized with weights that have been obtained from a pretrained network on similar data type (Zhuang et al., 2020). However, such approaches require a large initial dataset to pre-train the networks in the first place. Recently, (Billot et al., 2020a) had introduced a contrast agnostic learning strategy for MRI segmentation (Billot et al., 2020b). Their approach did not require any real data for training, as the synthetic based contrast images, and their corresponding segmentation maps, were generated "on the fly" during training. The generative process had been designed on a thorough physics driven understanding of the underlying processes that govern image contrast, which include effects such as the bias field and partial volume effects. Generic image segmentations maps were used as spatial priors, into which randomly varying gaussian mixture model based noise was added based on generic signal intensity characteristics for each segment.

In this work, we build upon the aforementioned work by considering a quantitative MRI based regression task. Unlike segmentation based neural network training, which, for the most part, requires pairs of single channel signal and segmentation images, the quantitative domain can often comprise of multi-channel input signals, that are governed by an underlying physics driven forward model. The forward model itself can rely on multiple independent effects, that arise from different sources. For example, in MWF based parameter mapping, the T2 signal decay is governed both by the underlying biophysical tissue properties, as well as the B1 transmit field inhomogeneity, both of which need to accounted for when building training data.

Furthermore, in quantitative MRI there also exists a need to synthesize signals that vary not only in the spatial domain, but also in the parametric domain. In other words, the underlying parameters of the forward model need to cover a wide variety of possible combinations of values, such that any possible MRI signals can be generated from it. Such generative models, which we will refer to as parametric models, have been previously demonstrated for synthesis of signals to train a voxelwise 1D networks (Barbieri et al., 2020; Yu et al., 2021). Therefore, in this work we introduce a framework for combining the contributions of spatial generative models, and parametric generative 1D models, to synthesize 3D datasets. We demonstrate how such a generative model can be applied to the specific problem of estimation of myelin water fraction maps from multi-echo T2 weighted CPMG based acquisitions.

## 2. Method

### 2.1. Signal Decay Model

A conventional way to quantify myelin is to derive its fraction from the probability distribution of $T_2$ decay rates, $p(T_2)$. A discrete parametric model of $p(T_2)$ is given by:

$$p(T_2) = \sum_i^n v_i \Psi_i (\mathbf{p}_i, T_2) \tag{1}$$

where $i$ is a component of the distribution, $v_i$ is the volume fraction, $\Psi$ is a density function, $\mathbf{p_i}$ are the parameters of this density, and $T_2$ is the decay rate. Among a large variety of parametric distribution models (Akhondi-Asl et al., 2014; Chatterjee et al., 2018; Raj et al., 2014), we choose the most commonly used one, where $\Psi$ is a Gaussian Mixture Model (GMM), with parameters $\{\mu_i, \sigma_i\}$, and three components $i \in \{1, 2, 3\}$, which represent myelin bound water, intra-extra axonal space (IES) bound water and cerebro-spinal fluid (CSF) free water respectively. The goal of myelin mapping is to recover the myelin bound water volume fraction (MWF), $v_m$, which is conventionally measured over $T_2$ range of 10ms to 40-60ms from $p(T_2)$ (Kolind et al., 2009; Whittall et al., 1997). Given $p(T_2)$, the observed MRI signal is calculated as:

$$I(TE_i) = \int \mathrm{GEC}(TE_i, T_1, T_2, \alpha) \, p(T_2) \, \mathrm{d}T_2 \tag{2}$$

where $TE$ is the echo time, $\alpha$ is the refocussing pulse flip angle, $T_1$ is a relaxation constant and GEC is the generalized echo curve model (Hennig, 1991). $TE$ is set by the specific acquisition sequence chosen at scan time. $\alpha$ will vary spatially from scan to scan due to B1 transmit inhomogeneities (Akhondi-Asl et al., 2016). $T_1$ is conventionally set to a pre-estimated value from literature to relax the calculation burden (Prasloski et al., 2012). Figure A.1 shows an example of the MRI signal and the corresponding $p(T_2)$ distribution for a single voxel.

### 2.2. Signal Decay Model Sampling

The generative model for sampling the parameters of the multi-component T2 signal decay model of $p(T_2)$ consists of 3 variables that are sampled independently from their individual priors - the three volume fractions $v_i$. To synthesize the volume fractions within the spatial constraints, we sample each from a uniform distribution, $P_{k,v_i} = U(u_{k,v_i}, l_{k,v_i})$, where $k \in \{1, ..., K\}$ is a brain segmentation region, $U$ is a uniform distribution, $u$ is an upper limit and $l$ is a lower limit. To choose a proper distribution of all possible $v_i$ for each region, we consider a separately acquired dataset with the same acquisition settings as other in-vivo scans, from which we derive the min-max variance of $v_i$ based on NNLS based estimates. These estimates are derived from a fine segmentation mask based on 285 unique regions of brain parcels and tissues (Hammers et al., 2003). It is important to note here that the model does not rely on NNLS estimates for training the neural network, but only for building a lookup table to determine the range of min-max values in each of the 285 segments, from which the values will be sampled. Finally, we also note that to constrain the search space of all possible parameters that we simulate in the synthetic data, we fix the $\sigma_i$ and $\mu_i$ of

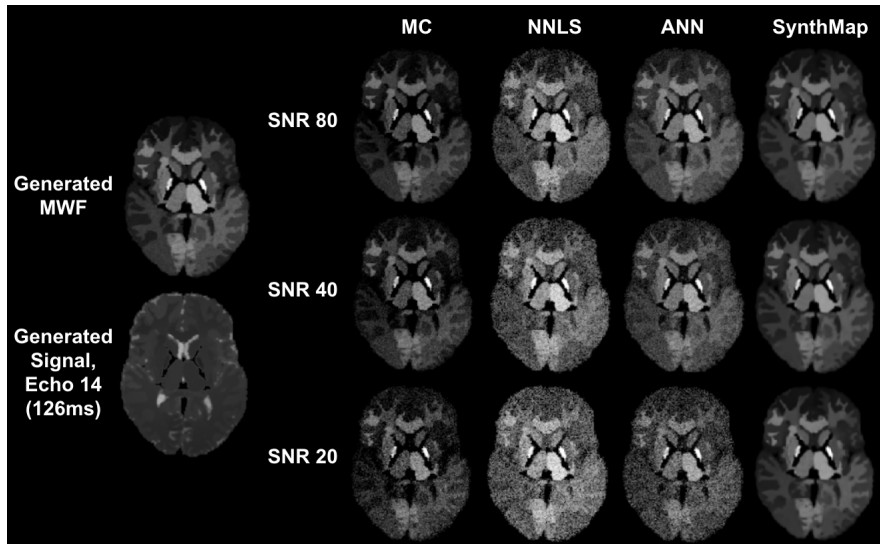

Figure 1: Synthetically generated MWF map and the corresponding generated signal. Estimated MWF maps for each method are shown at different SNR degradation levels. SynthMap consistently recovers smooth non noisy estimate of MWF.

each of the three components to constant values based on their narrow variance reported in literature (Chatterjee et al., 2018).

### 2.3. Spatial Model

Let $S$ represent a segmentation with $K$ labels, which we obtain independently from a generic database. We pose this segmentation as a prior anatomical distribution $p(S)$ and formulate a sampling process for a likelihood $p(I \mid S)$, to obtain a sample of an image $I$. The likelihood is modelled as a spatial GMM sampled at each voxel $j$ with parameters $\{\mu_k, \sigma_k\}$. By grouping these parameters into $\theta_G = \{\mu_1, \sigma_1, ..., \mu_K, \sigma_K\}$ the likelihood can written as a product over all $j$:

$$p(I \mid S, \theta_G) = \prod_j \mathcal{N}\left(I_j; \mu_{S_j}, \sigma_{S_j}^2\right) \tag{3}$$

Crucially, this model must be extended to a multi-variate case to accommodate for the signal decay model parameters dimension $v_i$. In this instance these parameters are described by $\{\mu_{k,v_i}, \sigma_{k,v_i}\}$. Augmentation is added to emulate anatomical variation and spatially varying physics driven processes induced within the MRI scanner. This is achieved by a multiplication of independently sampled spatial transforms - an affine matrix $\phi$, a non rigid deformation field $\tau_j$, and a blurring kernel $B$ to simulate partial volume effects. Finally, we must also obtain an independent sample of the flip angle of the refocussing pulse, $\alpha_j$, which simulates spatially varying B1 transmit inhomogeneity, which is necessary for the calculation of the MR signal in Equation 2.

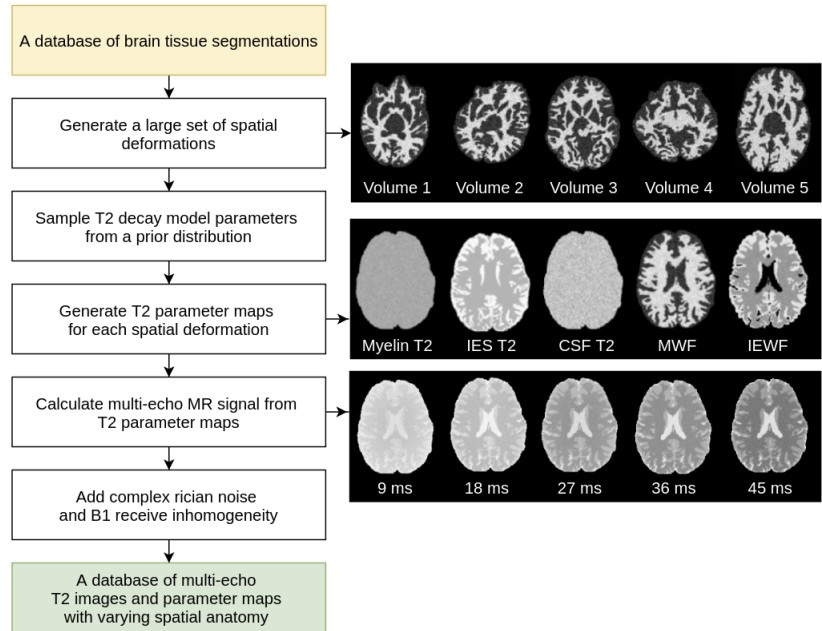

Figure 2: The proposed model consists of spatial generative model and signal decay generative model, which are combined together as shown in the flowchat. Each model's parameters are sampled independently over a wide range of prior values to represent variation in anatomy, scanner induced effects and the biophysical tissue model. Intermediate outputs of the generative model outputs are shown on the right.

## 2.4. Spatial Model Sampling

The matrix of $\phi$ consists of the independent components of three rotations, three scalings, and three shearings, which are all sampled from their respective predefined ranges of continuous uniform distributions. $\tau_j$ is built on the model of a smooth random stationary velocity field (SVF), and the sampling process is implemented as shown in (Billot et al., 2020a). $B$ is also sampled from a continuous uniform distribution to set the correct kernel size. $\alpha_j$ is obtained from a smooth spatially varying 3D spatial GMM, whose parameters are sampled independently for each of the axis from a continuous uniform distribution. It is important to note that the goal of this model is not to precisely reproduce the appearance of real MRI images, but to expose the neural network to a large variety of signals, which would improve its robustness and generalization capacity (Eugenio Iglesias et al., 2020). The diagram of the proposed model, with an example of intermediate outputs, is shown in Figure 2.

## 2.5. Synthetic and Volunteer Data

We synthesized 2,000 volumes of size 192x160x256 with the proposed generative model, with 2 channels in each volume, which represented $v_m$, $v_{ies}$ of the parameter model. $v_{csf}$ was calculated analytically, since the volume fractions must add up to a unit value. The

*GEC* parameters in Equation 2 were pre-computed as a dictionary with an increment of 0.25 degrees for each value of $\alpha$, using the Extended Phase Graph algorithm (Hennig et al., 2004), and used as a look up table during synthesis to shorten the computation time and the load on the GPU. Additionally, complex rician noise and a smoothly varying B1 receive inhomogeneity are added to the final images, as detailed in (Billot et al., 2020a).

## 2.6. Network Architecture & Training

Parameter estimation network was built with a U-net (Ronneberger et al., 2015) motivated by its robust performance and widespread use. Network architecture included: 4 resolution blocks, 3 convolutional (CNN) layers per block with a ReLU activation function & a dropout of 50% after every CNN layer, 64 features maps in the first CNN layer, a 3x3 CNN kernel, and a max pooling layer for each block. The last CNN layer output had 1 channel, which corresponded to $v_m$ estimate. Training parameters were: Adam optimizer (Kingma and Ba, 2014) with 0.0006 learning rate, and a batch of 32. Input consisted of 192x160x32 tensors with 32 echoes in the channel dimension and the output consisted of 192x160x1 myelin fraction maps, supervised with the L2 mean squared error loss. Parameter estimation network was trained on a database of 120,000 synthetically generated slices.

## 3. Results

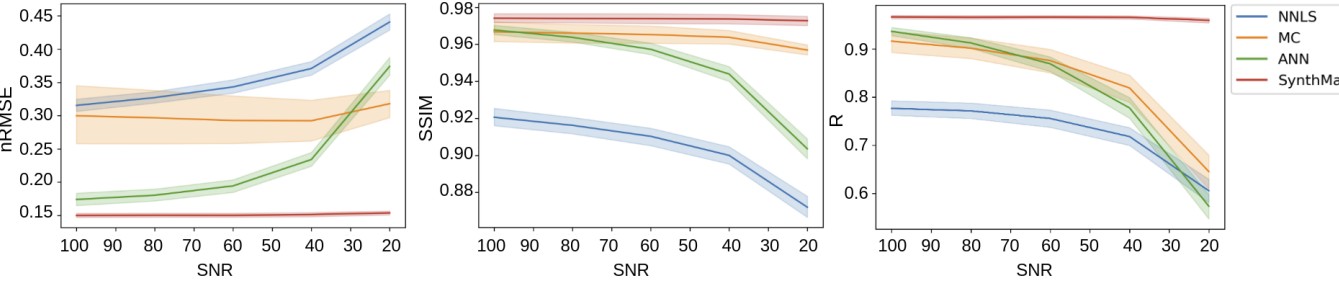

Figure 3: Quantitative comparison metrics between the ground truth MWF and the estimated MWF parameters on a hold out test set of the generated data, evaluated at different levels of rician noise. SynthMap produces stable results even at low SNR.

We evaluated our proposed approach (SynthMap), against three competing methods - 1. a widely used conventional NNLS based voxelwise scheme that estimates $p(T_2)$ as a linear combination of 40 impulse functions, from which $v_m$ is then derived (Doucette et al., 2020) (NNLS); 2. a multi-compartment probability distribution model of T2 values. This model uses a mixture of three Gamma distributions for which the mixture fraction parameters are estimated directly based on a variable projection optimization technique, and which correspond to myelin water, intra-/extra- axonal water and CSF water respectively (MC); 3. a 1D trained (voxelwise) supervised fully connected neural network, for which the myelin water fractions and the corresponding signals were synthesized with a dataset of 1.4Million

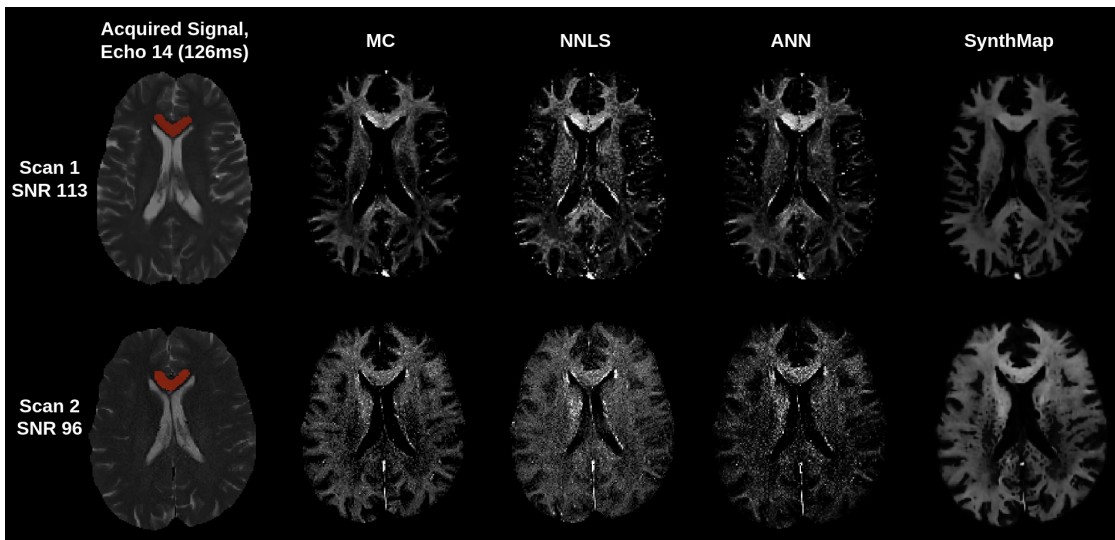

Figure 4: Two CPMG scans acquired at different resolution levels with the SNR of 113 and 96 respectively. Fitted myelin water fraction maps are shown with corresponding signal. Genu of Corpus Callosum is segmented in red over which mean MWF was measured for each method (see Table 1). High resolution scan 2 yields substantially degraded quality of estimated MWF maps in competing methods, while SynthMap is able to recover smooth details of the tissue.

| Scan # | Resolution mm$^2$ | SNR | Myelin Water Fraction mean (std) | | | |
|---|---|---|---|---|---|---|
| | | | MC | NNLS | ANN | SynthMap |
| 1 | 1.04x1.04 | 113 | 38 (17) | 48 (21) | 38 (22) | 36 (6) |
| 2 | 0.62x0.62 | 96 | 35 (15) | 48 (16) | 39 (21) | 35 (5) |

Table 1: Mean and standard deviation of the estimated myelin water fraction in the genu of corpus callosum (see reference in Figure 4) for each competing method for two different scans with different inplane resolution. SynthMap is able to recover the mean MWF in a similar manner as MC and ANN for both scans, while yielding a much lower variance.

voxels in a 1D sense (Yu et al., 2021) (ANN). Performance was evaluated on synthetic data first to measure accuracy under controlled conditions with the available ground truth myelin fraction maps. Figure 1 shows example myelin fraction maps estimated with SynthMap and the competing methods at three difference SNR levels. SynthMap exhibits sharp and realistic myelin fraction maps, while achieving a significant improvement in robustness to noise. NNLS and ANN models tend to produce noisy unstructured spatial information at SNR 40 or lower. A test set of 117 slices selected at random from 100 synthetic volumes was used on synthetic data to report nRMSE, SSIM and pearson correlation coefficient (R),

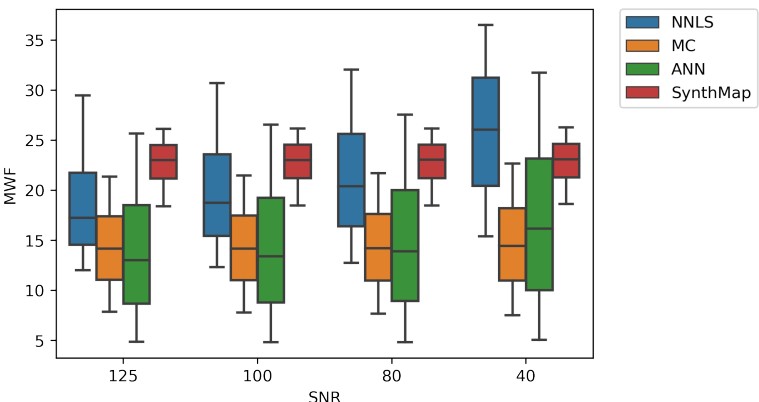

Figure 5: Aggregate estimate of MWF across the entire white matter across all healthy volunteer scans. Higher MWF for SynthMap may be attributed to denser map of myelin, as shown in an example in Figure 4.

as shown in Figure 3. Evaluation was also performed on volunteer data, by considering two individual scans at different inplane resolutions, which naturally reduced the SNR from 113 to 96 for each of the scans. Figure 4 shows that SynthMap is successfully able to recover the finer structural details on real data at lower SNR, and Table 1 shows a quantitative comparison between the methods. SynthMap is able to successfully recover parameters, similarly to MC and ANN, while preserving structural cohesion in the image. Aggregate measure of MWF across the entire white matter (WM) was taken from 7 healthy volunteer subjects, to verify the distribution of parameters, as shown in Figure 5. SynthMap on average yields higher values of MWF, when we consider the entire WM space. We refer the reader to Figure 1, which shows that each of the competing methods tends to underfit the true MWF values, which leads to a lower overall range of values, and higher variance (e.g. see ANN in Figure 5).

## 4. Discussion and Conclusions

This work proposed to combine a MR physics signal decay model, an accurate probabilistic multi-component parametric T2 model and a contrast agnostic spatial generative model to improve the estimation of T2 distributions from multi-echo T2 data. Our generative model is able to create a large synthetic dataset (e.g. 120,000 slices) that could be used to efficiently train deep neural networks directly, without the need for real data, to output the parameters of the underlying T2 distribution. This is beneficial for MWF estimation, as the availability of high quality MWF maps is scarce, due to substantial scanning time that is required to acquire volunteer data, lack of clinical scan protocols, and high sensitivity even to small decreases in SNR (as shown in our experiments). The model is robust to noise, which is crucial for adoption of myelin mapping techniques in clinical practice that may be achieved with accelerated imaging (Deshmane et al., 2012) to reduce scan time.

## Acknowledgements

This work was supported partially by the NIDDK, NIBIB, NINDS and NLM of the National Institutes of Health under award numbers R01DK125561, R21DK123569, R21EB029627, R01NS121657, R01LM013608, S10OD0250111 and by the grant number 2019056 from the United States-Israel Binational Science Foundation (BSF), and a pilot grant from National Multiple Sclerosis Society under Award Number PP-1905-34002.

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

## Appendix A. Signal Decay Model

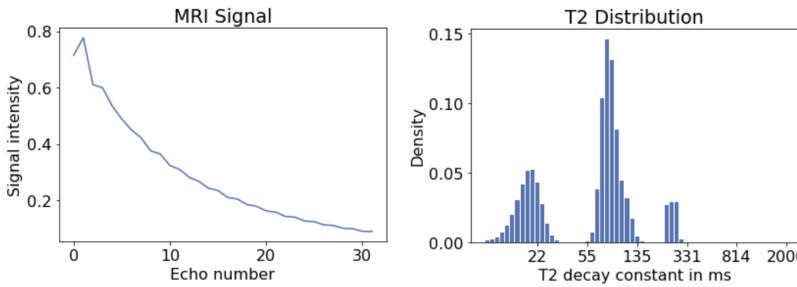

Figure A.1: A single voxel example of the MRI signal decay (left) and the corresponding T2 distribution (right) generated with the proposed model. MRI signal decay (left) has a distinct peak on the second echo due to B1 transmit effects. Three components of p(T2) (right) represent myelin, intra-/-extra axonal space and cerebro-spinal fluid respectively.

## Appendix B. In-vivo Data Acquisition

We performed an IRB approved and HIPAA compliant study to image 7 healthy volunteers. The data were acquired on a 3D Siemens Skyra scanner with the following sequences: a) 2D multi-echo CPMG sequence with 1 mm$^2$ in plane resolution, 4mm slice thickness, 32 equally spaced echoes covering a $TE$ range of 9-288ms, TR of 2000ms, 24 slices, acquired in 9 minutes with 5/8 Partial Fourier and in-plane GRAPPA with an acceleration factor of 2. The acquisition was performed in slabs to limit magnetization transfer effects, with 4 concatenations and 5 slice gap between consecutive slices b) a reference high resolution T1-weighted MPRAGE scan - 220x220 matrix, 176 slices, 1x1x1 mm3 resolution, 2.27ms TE, 1100ms T1, 2000ms TR b) a reference high resolution 3D T2-weighted scan - 256x256 matrix, 160 slices, 0.99 mm thickness, 3200ms TR, 441m TE.

## Appendix C. Permutation Study

The goal of this experiment was to evaluate whether the pretrained CNN based network is learning to resolve the non-linear relationship in the time domain that is present in the signal decay curve for each voxel, rather than learning to simply transfer the MWF values that correspond to a specific anatomical region in the synthetic training data to the in-vivo test data. To do this, five 5x5 patches were marked in anatomically distinct regions of the brain which are expected to have different MWF estimates - Corpus Callosum, Frontal White Matter, Occipital White Matter, Major Forceps, Cortical Gray Matter area. Next, 10 different permutations were designed, whereby the patches were swapped with each other. The resulting 10 permutations of the input image, alongside with the original unperturbed image, were fed through the synthetically pretrained CNN network. Figure C.1 shows an example estimate of the MWF with the pretrained Synthmap when two distinctly different

regions of the same slice were swapped together. Notably, the CNN was able to recover the MWF values correctly, as shown by the dark region in the Corpus Callosum, and the bright region in the Cortical Gray Matter region. Table 2 shows a measure of the coefficient of variation (CoV) of the mean MWF from each patch, as it was placed in different regions of the image. Results indicate that even if a specific patch was located in an area that was distinctly different to its original location, with very different MWF values of the tissue around it, the pretrained network had predicted the values correctly. We note that some variation in the mean MWF from each patch is expected, as the CNN is applying convolutional kernels that will smooth the edges of the kernel with the surrounding tissue. The measured CoV is highest for the Genu of the Corpus Callosum patch, as its values are typically the highest in the entire brain, and therefore its placement in other regions, with lower MWF, would cause the patch edge smoothing effect to have the largest impact.

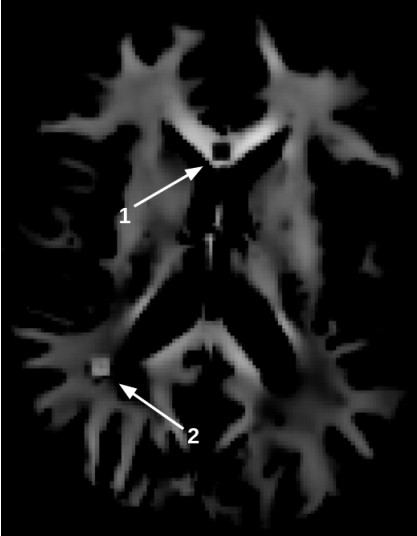

Figure C.1:  Estimate of the MWF map by the synthetically pre-trained Synthmap on an in-vivo volunteer data where the input signal from two distinctly different regions of the same slice were swapped. This result indicates that the network correctly predicts values based on the signal decay properties of each voxel.

## Appendix D.  Synthetic Data Pre-Processing

Synthetized 3D volumes were split into training, validation, test data. Each set was then split into individual 2D slices. Consequently, slices from a specific volume were exclusive either to training, validation or test set.

Upper and lower anatomy of the brain was limited to match the areas of the brain that were covered by the in-vivo volunteer scans, which reduced the effective number of slices taken from each synthetically generated volume. Included regions consisted of - all of the anatomy above the midbrain, which includes the Corpus Callosum in its entirety, as well as the Thalamus, Putamen, Caudate Nucleus, Minor Forceps, Major Forceps; as well

| Patch Location | Mean MWF at original location | Std across permutations | CoV across permutations (%) |
|---|---|---|---|
| Genu of Corpus Callosum | 31.5 | 2.1 | 6.7 |
| Cortical Gray Matter | 21.2 | 0.8 | 3.8 |
| Frontal White Matter | 27.8 | 1.3 | 4.7 |
| Major Forceps | 19.8 | 0.7 | 3.5 |
| Occipital White Matter | 18.4 | 0.9 | 4.9 |

Table 2: Mean MWF of a specific patch in its original (true) location is presented in column one. Standard deviation of the mean over different permutations of locations (as the patch is moved to different location in the brain) is given in column two. Coefficient of Variation (CoV) over the permutaitons is given in column three. Low CoV indicates that the network correctly predicts MWF in each patch, even if it is moved to a different location. Some variation in MWF is expected due to smoothing properties of the CNN kernels.

all of the area that is below and including the Cingulate Gyrus. Example of regions that were excluded were: Pons, Medulla, Cerebellum, as well as very smaller slices (small brain footprint) in the upper regions of the cortex.

We note that the synthetic training data was arranged as 2D slices, rather than 3D volumes because the in-vivo acquired data is in the form of slabs composed of 2D slices with a slice gap in between slabs, and therefore slabs cannot always be perfectly aligned into a consistent 3D volume. For alternative acquisition sequences, such as the 3D GRASE, a 3D input volume can be arranged with a 3D network architecture. The use of this 3D consistency may yield an improved performance. However, there will be specific challenges associated with this kind of design - 1) 4D volume inputs of size 192x160x24x32, where 32 refers to the time decaying echo signal, may require a patch based 3D architecture due to the GPU memory input constraints 2) 3D sequences, such as the GRASE sequence, would often yield motion artifacts that appear across the entire image, and may be an additional challenge, particularly for those methods that consider spatial information, and not just individual voxels.

## Appendix E. ANN method

The ANN method was evaluated on the Tensorflow based implementation of the original paper by (Yu et al., 2021), which is made available by the authors in this Github repository - https://github.com/thomas-yu-epfl/Model_Informed_Machine_Learning. The training scheme was fixed to the details provided in the original code, including all of the hyperparameters, such as the optimizer and its learning rate, the weights initializer, batch size and the number of predicted impulse functions. Since no early stopping criteria was provided, we implemented a patience of 10 epochs over the performance of the validation loss as the automatic stopping criteria. Furthermore, the network was re-trained multiple times with different random fixed seeds, to ensure the best possible performance on the training data. The training was performed on the same hardware as the Synthmap.

Importantly, the ANN predicts a T2 distribution of 60 impulse functions as the output, rather than a direct estimate of MWF. Conventionally, to convert a T2 distribution to a MWF value, a cut off value is fixed to a measure of the area under the T2 distribution curve at 50ms. Such conversion approach was used both for NNLS and ANN methods. We note that this inherent dissimilarity between how a MWF is derived (a measure of a fraction of the area under the curve for a T2 distribution for ANN and NNLS; and a direct measure of MWF for MC and Synthmap) may yield some difference in the performance. Further detailed evaluation of this may be beneficial in the future works.

The ANN was trained based on the synthetically generated dataset of 1.4Million voxels that was provided by authors of the original work. Each pair of training points consisted of - 32 echoes that represented the signal decay curve as the input, and a T2 distribution of 60 impulse functions as the ground truth 'label'. The 1.4 Million dataset was equally split into 7 distinct tissue types, as outlined in the original paper. The representation of each region was balanced (200,000 voxels per region), which would ensure that the ANN is not biased to any particular tissue type during training, as the inputs are fed voxelwise. We note that while it may be possible to use our synthetically generated 3D data to train a fully connected network in the voxelwise manner, such as the one put forth by the authors of the ANN, the representation of specific tissue types would not be balanced in such training due to the anatomically different coverage of each region. Therefore a decision was made to use this 1D synthetic dataset for training the ANN for a fair comparison of performance. We further note that this synthetic dataset of 1.4Million voxels was designed, and optimized, specifically for evaluation of brain MWF on in-vivo data.

## Appendix F. NNLS and MC methods

The NNLS method was evaluated on the Julia language based implementation of the original paper by (Prasloski et al., 2012), which is made available under the DECAES software - https://github.com/jondeuce/DECAES.jl. The output consisted of a prediction of 40 impulse functions that represent the T2 distribution for the given signal decay of a voxel. MWF was derived from the T2 distribution by fixing a cut off value at 50ms, to calculate the fraction of the area under the curve.

MC method was evaluated on the C++ based implementation of the original paper by (Chatterjee et al., 2018), which is made available under the ANIMA software - https://github.com/Inria-Empenn/Anima-Public. The C++ pre-compiled binaries were wrapped into a docker container - https://github.com/sergeicu/anima-docker. The predicted output consisted of the direct measure of myelin fraction, as well as IE fraction and CSF fraction.

## Appendix G. Table 1 Supplement

In this section we present two additional regions of interest as a supplement to Table 1 in the main manuscript - Splenium of Corpus Callosum and Putamen. Estimated MWF is shown by the competing methods in two scans of different SNR levels. SynthMap is able to recover the mean MWF in a similar manner as MC for both scans, while yielding a much lower variance.

| Scan # | Resolution mm$^2$ | SNR | Myelin Water Fraction mean (std) | | | |
|---|---|---|---|---|---|---|
| | | | MC | NNLS | ANN | SynthMap |
| 1 | 1.04x1.04 | 113 | 21 (13) | 35 (24) | 16 (15) | 23 (3) |
| 2 | 0.62x0.62 | 96 | 25 (19) | 45 (17) | 29 (22) | 26 (4) |

Table 3: Mean and standard deviation of the estimated myelin water fraction in the Splenium of Corpus Callosum for each competing method for two different scans with different inplane resolution.

| Scan # | Resolution mm$^2$ | SNR | Myelin Water Fraction mean (std) | | | |
|---|---|---|---|---|---|---|
| | | | MC | NNLS | ANN | SynthMap |
| 1 | 1.04x1.04 | 113 | 26 (12) | 33 (15) | 16 (12) | 28 (3) |
| 2 | 0.62x0.62 | 96 | 32 (16) | 45 (18) | 31 (22) | 31 (4) |

Table 4: Mean and standard deviation of the estimated myelin water fraction in the Putamen for each competing method for two different scans with different inplane resolution.

