# OpenReview forum: "SynthMap: a generative model for synthesis of 3D datasets for quantitative MRI parameter mapping of myelin water fraction"
_MIDL.io/2022/Conference — MIDL 2022_

### Official Review · Reviewer_WJxE · 2022-01-22

**Confidence:** 3
**Preliminary Rating:** 4
**Recommendation:** Poster

**Summary:**

A generative model is developed for synthesizing datasets for quantitative mapping of myelin water fraction (MWF). The biophysical model that describes the relationship between the MRI signals and the underlying signals is leveraged for the synthesis, together with a generative spatial model to capture spatial variations.


**Strengths:**

- The use of a biophysical model allows integration of domain knowledge into deep learning.
- Spatial variation is also considered in the image synthesis.
- Both simulation data and real data were used for validation.

**Weaknesses:**

- The experimental setting for the competing method ANN needs to be better clarified.
- The results on the real data still needs to be better discussed. It is not clear how to determine the results are better or not.

**Deanonymize Review:**

no

**Detailed Comments:**

- Is a 2D CNN used for the estimation? Since the images are 3D, why not use a 3D U-Net? I am wondering which would be better, 2D or 3D U-net?

- Is the ANN used for comparison trained with the same data used by the proposed method? For fair comparison, I believe it should be.

- In Fig. 5, the result is shown for the aggregate measure of MWF. Is a higher value better? I am not sure what conclusion can be drawn from this figure.

**Final Rating After The Rebuttal:**

5: Strong Accept

**Justification Of The Final Rating:**

The authors have well explained the use of 2D CNN. The training configuration is also introduced for a fair comparison. The implications of the results are discussed. Based on the responses, I have upgraded my recommendation.

**Paper Type:**

methodological development

**Questions To Address In The Rebuttal:**

- Fair comparison should be ensured for the comparison between methods.
- The results on the real data can be better discussed. In particular, what does the higher MWF achieved by the proposed method imply?

**Special Issue:**

no

---

### Official Review · Reviewer_LzFU · 2022-01-24

**Confidence:** 4
**Preliminary Rating:** 3
**Recommendation:** Poster

**Summary:**

The authors propose a generative model for the creation of Synthetic parametric maps of multi-echo T2 images and use it to train a regression network to estimate the spatial map of myelin fraction. The proposed model is compared on synthetic and real data against 3 other existing models at different ranges of resolution and SNR. The authors show in particular the stability of the performance across a wide range of noise levels.

**Strengths:**

- Good motivation for the paper
- Detailed explanation of the synthetic image generation
- Attempt to perform the evaluation on both synthetic and real data
- Comparison performed also at different levels of image quality

**Weaknesses:**

- Some of the section titles don't reflect well the content of the paragraph (2.2 does not say anything about volunteer data)
- Possibly unfair comparison as the evaluation is performed on images simulated with the same paradigm as the training images (at least for the methods requiring training)
- Lack of clarity of the experimental design related to volunteer data as to the way the ground truth is obtained
- Everything appears to be done on 2D and there is no discussion on the impact/the need of 3D consistency

**Deanonymize Review:**

no

**Detailed Comments:**

Given that 2000 images were generated amounting to 512000 slices, could the authors explain why only 120 000 slices were used for the training?
It is not clear to what $\mu_k$ and $\sigma_k$ refer to in the description of the spatial model.

**Paper Type:**

methodological development

**Questions To Address In The Rebuttal:**

It would be good to particularly address the comments related to
- the clarification of the spatial model
-  the interrogations related to the fairness of the comparison
-  the details of the use of volunteer data.

**Special Issue:**

no

---

### Official Review · Reviewer_B1So · 2022-01-24

**Confidence:** 4
**Preliminary Rating:** 3
**Recommendation:** Poster

**Summary:**

The authors present a method named SynthMap that combines the synthesis of 3D anatomical datasets and MR physics-based signal evolutions. This allows training spatially constrained CNN models for MWF estimation based on T2 signal decay measurements.

Instead of estimating quantitative parameters using conventional voxelwise signal fitting algorithms, the authors propose to train a CNN network (U-Net architecture) for joint regularization in the spatial and parametric domain. With their SynthMap approach, they present a solution to overcome the usual bottleneck in such supervised learning tasks, which is the lack of paired training data.


**Strengths:**

Overall, the paper is well written and clearly structured.

The paper presents a solid data synthesis methodology. It is well described how the anatomical context information is fused with the MR physics-based signal modelling.

The authors claim convincingly how the use of physics inspired data synthesis can help to solve the need for high-quality training data and propose to close this research gap by generating a self-contained, paired dataset for DL model training and to evaluate model performance afterwards.

Performance evaluation also comprises evaluation with reference methods, i.e., NNLS, MC and ANN approaches.
The results are promising - the CNN-based spatial regularization results in a clear improvement compared to single-voxel approaches with respect to spatial consistency.

The authors released their source code.


**Weaknesses:**

Although the results look promising, there is little insight on how the CNN model is trained, validated and tested. The following details would benefit from further justification:
-	In particular, how was the generated dataset split? Was training, validation, test data split anatomy/volume-wise?
-	Why was only such a small portion of the 2,000 volumes (à 250 slices each) used for training (120,000 slices)?
-	Regarding the experiment in Fig.1: Did the training data comprise all the different noise levels or was the noise level used to corrupt the synthetic data adapted to the tested noise level?
-	In a similar vein, it is not fully clear to me if the ANN was trained on the same synthetic data, just on a voxel-wise manner or if a separate set of signal curves was simulated.

Indeed, the MWF obtained with SynthMap look visually more appealing than the voxelwise approaches. However, I am wondering if the CNN might has only learnt to transfer the underlying brain anatomy instead of resolving the non-linear relationship of the WMF and the acquired signal-time curve (which is a priori a 1D problem).

I am also missing details about the in vivo MR acquisition, e.g., MR sequence scheme, scan parameters etc.
Quantitative analysis of the in vivo data is limited to one specific WM region, i.e., the genu of the corpus callosum. I am wondering why?

It would certainly increase the quality of the paper if the authors also touched upon the limitations of their work.


**Deanonymize Review:**

no

**Detailed Comments:**

-	“Complex rician noise”: Can you please clarify if you apply Gaussian noise to the complex MRI signal or Rician noise to the magnitude signal?
-	“[..] we consider a separately acquired dataset, from which we derive the min-max variance […]”: What kind of dataset is it? Same acquisition scheme as for the other in vivo measurements?
-	Page 6: “the first CNN layers 3x3 CNN kernel”
-	Page 6: “the output consisted of 192x160x1 myelin fraction maps”


**Final Rating After The Rebuttal:**

4: Weak Accept

**Justification Of The Final Rating:**

The proposed SynthMap approach is a meaningful advance to the research field and fits well into the scope of MIDL. My remaining questions and suggestions are now well incorporated in the revised manuscript or have been clarified in the authors’ reply. I really like the permutation experiment! I am happy to update my rating.

**Paper Type:**

both

**Questions To Address In The Rebuttal:**

I would want the authors to expand on the aspects discussed under weaknesses during the rebuttal period.
It would be particularly important to better understand the training/evaluation setup and to reflect on the limitations of the work.

**Special Issue:**

no

---

### Meta-Review · Area_Chair_wWcw · 2022-02-16

**Recommendation:** Accept (Poster)
**Confidence:** 4

**Metareview:**

The presentation, organization, and writing of this paper is good. The data synthesis methodology is solid and convincing. The experimental results are promising. The implementation of this work is publicly available. One reviewer vote for borderline, and two reviewer recommend accept. The rebuttal addresses the concerns from all reviewers.

---

### Decision · Program_Chairs · 2022-02-28

Accept